# Financial market regulation and corporate social responsibility: Evidence from China's new asset management regulation

Le Zhu[1], Yichuan Wang[2*], Quan Zhang[3]

**1** School of Accounting, Shandong Technology and Business University, Yantai, China, **2** W. A. Franke College of Business, Northern Arizona University, Flagstaff, Arizona, United States of America, **3** School of Accounting, Shandong University of Finance and Economics, Jinan, China

* yichuan.wang@nau.edu

## Abstract

This study explores how the financial market environment reshapes corporate social responsibility using a quasi-natural experiment provided by China's New Asset Management Regulation. Our research focuses on the adaptive strategies of non-financial firms in response to stringent financial market regulation, and we use a generalized DID model to identify the causal link between the NAMR and CSR. The findings reveal a decline in non-financial firms' CSR performance following the more stringent financial market regulation. Mechanism testing suggests that the negative impact is primarily due to the reduction in the return on financial asset investments. Furthermore, we assess the heterogeneity influences of financial regulation on the three dimensions of CSR (environment, society, and governance). Our analyses underline a significant decrease in the environment and governance CSR among non-financial firms, while no significant impact is observed on the social dimension of CSR. This study contributes to a greater understanding of the relationship between financial market regulation and CSR. It offers valuable insights for the development of effective policy guidance to ensure the optimal functioning of the real economy.

## Introduction

Financial liberalization, deregulation, and excessive speculation have long been recognized as key drivers of financial crises. In response to the 2008 global financial crisis, governments worldwide have placed an unprecedented emphasis on financial market regulation. At the macroeconomic level, financial regulation has been shown to reduce bank risk [1], enhance national financial stability [2], and promote economic growth [3]. However, excessive regulation poses risks, including stifling the growth of certain financial systems and undermining competitive equity [4]. Additionally, overregulation may weaken the positive impact of inclusive finance on economic growth [5].

**Data availability statement:** The data underlying the results presented in the study are available from (China Stock Market & Accounting Research https://data.csmar.com/).

**Funding:** This work was supported by the Key Program of National Social Science Foundation of China under Grant [No. 20AGL010]. The funders had no role in study design, data collection and analysis, decision to publish, or preparation of the manuscript.

**Competing interests:** The authors have declared that no competing interests exist.

At the microeconomic level, robust financial regulation has been argued to reduce managerial agency costs and mitigate risk aversion, fostering a greater willingness to undertake risky investments [6], ultimately improving resource allocation efficiency [7]. Nevertheless, a "one-size-fits-all" regulatory approach may impose substantial economic costs. For instance, Jiang et al. found that while the new asset management regulation significantly curtailed shadow banking activities, it also increased financing costs and constraints for real-economy enterprises [8]. Similarly, Liu reported that firms with higher levels of financialization before the regulations responded by raising labor costs [9].

Given that financial markets serve as a crucial platform for firms to raise capital and make investments, changes in these markets inevitably shape both their external environment and internal resources, influencing investment decisions. Previous research has shown that stricter financial regulation can encourage firms to invest in R&D [10] and improve asset investment efficiency [11]. However, unlike R&D and other tangible assets, CSR investments have longer payback periods and present challenges in measuring returns. While extensive literature explores the drivers of CSR, little attention has been given to the impact of financial market regulation on CSR. Corporate social responsibility (CSR) plays a crucial role in corporate operations. As a risk management tool, it mitigates adverse events and reduces stock price collapse risk, functioning as an insurance-like mechanism [12–15]. As a value investment, CSR fosters stakeholder trust [16,17], boosts investor confidence [18], and strengthens political ties [19], ultimately enhancing business performance and firm value [20]. In financial capitalism, the financial market serves as both a platform for governments to promote CSR engagement and a constraint that may limit firms' ability to invest in CSR [21]. Marti and Scherer argue that advancing corporate social justice and welfare requires government intervention in the financial sector [22]. However, how firms navigate CSR investments in a more regulated financial market remains an open question—one that this study aims to address.

China's regulatory framework for the asset management of financial business provides a unique opportunity for our investigation. In April 2018, China implemented the 'Guiding Opinions of PBC, CBIRC, CSRC, and SAFE on Regulating the Asset Management Business of Financial Institutions' (abbreviated as the NAMR), marking a significant shift in asset management regulation.

The NAMR aims to mitigate financial risks, redirect capital flows to the real economy, and support economic restructuring. Key provisions include: Elimination of Rigid Redemption and Fund Pooling: Prohibits financial institutions from guaranteeing returns or using rolling issuances for implicit investor protections. Fund pooling, which causes maturity mismatches and opacity, is explicitly banned.

Net Worth Management: Requires asset management products to reflect actual returns and risks, moving away from the traditional "capital preservation and guaranteed returns" model.

Investor Classification and Suitability: Categorizes products into public and private placements and raises qualification thresholds based on asset size and investment experience.

Standardized Asset Rules: Clarifies criteria for standardized debt assets and sets strict limits on the investment ratios and durations of non-standard assets to manage liquidity risks.

Leverage Restrictions: Caps leverage ratios, limiting public placement products to 140% of net assets and closed-end products to 200%.

Enhanced Risk Management and Disclosure: Mandates stricter liquidity management, addresses maturity mismatches, and enforces transparent fund investment and risk disclosure.

Independent Subsidiaries and Custodial Requirements: Encourages financial institutions to establish independent asset management subsidiaries to segregate risks, while product custody is mandated to be handled by qualified third parties.

Prohibition of Channel Businesses and Nested Investments: Bans channel services bypassing regulatory requirements and restricts nested investments to prevent regulatory arbitrage.

Unified Standards and Penetrative Regulation: Ensures consistent asset management standards, clarifies underlying assets, and identifies ultimate investors to improve transparency.

Implemented concurrently nationwide without regional trials, the NAMR represents the most comprehensive financial regulation in China's history. The policy aimed to standardize the asset management industry across the entire country to mitigate systemic financial risks. The implementation began immediately upon its release in April 2018, with a transition period extending until the end of 2020 to allow financial institutions sufficient time to comply with the new regulations.

Institutionally, the NAMR provides a robust framework for the standardized development of the asset management sector. Strategically, it optimizes resource allocation, enhances risk management, and strengthens the stability and sustainability of China's financial system. Furthermore, the NAMR serves as an exogenous shock to firms, as it was an externally imposed regulation enforced uniformly across the country without prior trials or gradual phase-ins. This meant firms had no opportunity to anticipate or adjust for its impact, making it independent of individual firm behavior.

The NAMR has two main effects for most firms. First, it raises entry barriers for engaging in shadow banking, reducing expected returns from financialized assets [10]. Second, it prohibits financial institutions from participating in cash-pooling businesses involving rolling issuance, aggregate operations, separate pricing, and rigid payments. These policies reinstate corporations as entities that bear the risk and reduce the incentives for them to engage in shadow banking operations. Building on this framework, we treat the NAMR as an exogenous shock and employ a generalized Difference-in-Differences (DID) model to analyze whether non-financial firms increase or decrease their CSR investments in a more stringent financial regulatory environment.

The study makes two significant contributions. First, it advances our understanding of the microeconomic consequences of stringent financial market regulation. While prior research has largely focused on financial regulation's effects on monetary policy transmission, shadow banking, and corporate investment behavior, limited attention has been given to its influence on firms' strategic decision-making in corporate social responsibility (CSR). Using the NAMR as a quasi-natural experiment, we empirically examine how stricter financial market regulations reshape firms' non-financial investment priorities. Our findings provide micro-level evidence on the unintended consequences of financial regulation, demonstrating that when faced with increased financial constraints, firms tend to de-prioritize CSR investments. This highlights the trade-offs firms encounter between financial stability and social responsibility, contributing to broader discussions on the economic impact of regulatory interventions. Second, our study expands the scope of financial market research by shifting the focus from traditional investor- and market-centric perspectives to firm-level strategic responses. Existing literature predominantly examines investor sentiment, market efficiency, and capital allocation under regulatory constraints while overlooking how financial regulations shape corporate decision-making beyond financial investments. A key feature of our study is the use of publicly available ESG scores to measure CSR outcomes, ensuring replicability and robustness. Our findings challenge the assumption that stronger financial regulations inherently lead to improved corporate behavior. Instead, we reveal a nuanced relationship where heightened financial constraints force firms to prioritize short-term financial stability over long-term social and environmental commitments. By integrating perspectives from

financial regulation, corporate strategy, and CSR, our study provides a comprehensive view of how firms adapt to evolving financial environments. Our findings have important implications for policymakers, investors, and corporate leaders in balancing regulatory objectives with broader societal goals.

## Literature review and research hypothesis

### CSR rationale: Shareholder value versus stakeholder value

The fundamental question in CSR research is, "What is corporate social responsibility?". Two dominant perspectives shape the field: one centered on maximizing shareholder value and the other on prioritizing stakeholder value. Understanding these perspectives is crucial for advancing CSR discourse and practice.

The shareholder value maximization perspective asserts that a firm's social responsibility is to enhance shareholder value. This view was articulated by Friedman in 1962, who argues that "The social responsibility of business is to increase its profits [23]" Building on this, Tuzzolino and Armandi argue that businesses must prioritize profit to ensure survival and growth, with social responsibility aligning with this goal [24]. Within this framework, corporate social initiatives are often viewed as an agency problem and a form of self-interested behavior by managers [25–28]. Extensive investments in CSR can lead to corporate resource inefficiencies and harm shareholders' interests due to information asymmetry [29,30].

In contrast, the stakeholder value maximization perspective views CSR as fulfilling the normative expectations of society's key groups. This view argues that socially responsible firm should balance the interests of all stakeholders, maximizing corporate value rather than solely prioritizing shareholder interests [31,32]. Corporate managers have a moral obligation to protect the legitimate rights and interests of all stakeholders [33]. Furthermore, Freeman and Velamuri emphasize that the primary goal of CSR is to create value for stakeholders [34].

According to the stakeholder theory, CSR has long-term effects, as stakeholders assess a firm's legitimacy based on its CSR performance, influencing stakeholders' decisions to provide critical resources [35–37]. Thus, fulfilling CSR is essential for enhancing a firm's stakeholder relationships. Proactive CSR engagement can help attract talented employees [38], enhance corporate reputation [39], gain consumer acceptance [40], and obtain more financing or strategic resources [42] leading to sustainable competitive advantages. Moreover, prioritizing stakeholder interests can increase their support for the firm, positively impacting shareholder wealth [41].

### Research hypotheses

The contrasting views on CSR lead firms to adopt different approaches in response to stricter financial market regulations, forming the basis for two competing hypotheses. From the shareholder value maximization perspective, firms may perceive CSR as a cost and reduce their CSR investments to conserve resources. Conversely, from the stakeholder value maximization perspective, firms may view CSR as an opportunity to enhance long-term value, driving a proactive commitment to CSR initiatives despite regulatory constraints.

**Shareholder value maximization: The negative effect of the NAMR on CSR.** Traditionally, the primary goal of corporate finance management has been to maximize shareholder value [42]. When regulatory arbitrage opportunities exist in the financial market, investing in financial assets tends to be more lucrative than investing in tangible assets. This incentivizes firms to prioritize financial investments for profit [43], leading to a continuous expansion in financial assets. As a result, financial investment returns become a significant profit source for firms [44]. The implementation of the NAMR is expected to diminish the returns on firms' financial asset investments. By explicitly prohibiting multi-layer nesting and imposing stringent oversight, the NAMR limits firms' ability to engage in shadow banking activities, thereby curtailing high-yield but high-risk investment opportunities. Additionally, after the removal of rigid payment, product issuance without credit endorsement reduces the risk-free rate of financial products and increases return volatility [45], further raising the investment risk associated with financial products. As a result, Firms may be less inclined to invest in financial assets, the

NAMR restricts businesses' participation in shadow banking, reduces the amount of financial asset allocations, raises the risk of investing in financial products, and has a negative impact on business earnings [10].

Due to resource constraints, business managers must often make trade-offs among various investment projects. When firm profitability declines, especially leading to deteriorating balance sheets, managers prioritize short-term, high-yield businesses to mask adverse performance signals [46]. As Edmans notes, "ESG is both extremely important and nothing special." "It's nothing special since it's no better or worse than other intangible assets that create long-term financial and social returns" [47]. Rooted in shareholder value maximization, managers under performance pressure may prioritize profit-driven investments while neglecting or abandoning CSR. Empirical evidence suggests that managers frequently weigh corporate investment against CSR, sometimes opting to exclude CSR projects [48–50]. Moreover, in the context of strict financial rules and increased financing obstacles, firms tend to take a risk-averse stance, considering CSR as an additional cost to be minimized in favor of profit maximization for shareholders. This leads us to **Hypothesis 1a**:

H1a: Based on the principle of maximizing shareholder value, stricter financial market regulation reduces non-financial firms' CSR.

**Stakeholder value maximization: The positive effect of the NAMR on CSR.** Unlike the "shareholder-first" approach, which prioritizes short-term profit maximization, CSR emphasizes long-term sustainable development and stakeholder value. Given resource constraints, firms prioritize maintaining solid relationships with shareholders, creditors, employees, consumers, suppliers, the government, and society to maximize stakeholder interests. This commitment involves prioritizing social performance over financial metrics, allocating resources towards product development and other industrial development activities to fortify core business competitiveness, which promotes long-term growth in firm value [51]. The implementation of the NAMR has reduced firms' inclination to invest in financial assets by restricting channels for participating in shadow banking and increasing the risk of financial products, which has redirected resources from the financial market towards social responsibility activities.

However, while the absence of financial market regulation amplifies financial risks, shadow banking and informal financing channels have historically provided some firms with crucial funding opportunities. The implementation of the NAMR has restricted informal financing channels, increased financing difficulties for firms. CSR plays a vital role in easing financing constraints by facilitating access to financial resources from banks, investors, and other stakeholders [52]. On the one hand, CSR can compensate for shortages in financial reports, reduce information asymmetry between banks and firms, and secure more credit capital support and extended loan terms for firms [53]. On the other hand, CSR serves as a signaling mechanism [54], bolstering investor confidence in firms' financial stability and social consciousness, thereby mitigating concerns surrounding future operational uncertainties and reducing equity capital costs [55,56]. Consequently, after implementing the NAMR, firms are likely to engage more actively in social responsibility activities to enhance their CSR performance, aiming to secure more credit financing conditions and alleviate financing constraints. Moreover, as China continues to promote green finance development, CSR performance has become an essential criterion for credit issuance by financial institutions, further incentivizing enterprises to fulfil their social responsibilities. Therefore, we propose **Hypothesis 1b**:

H1b: Based on the principle of maximizing stakeholder value, stricter financial market regulation enhances non-financial firms' CSR.

## Research design

### Data and sampling

Our sample consists of all publicly traded Chinese A-share non-financial firms between 2015 and 2022. CSR data is sourced from Bloomberg's ESG score rating data, while firms' financial data and governance information are from the

China Stock Market Accounting Research (CSMAR) database. Upon merging these datasets, we apply the following data processing steps: (1) Firms classified as ST or ST* are excluded. (2) Observations without data for at least one period before and after the NAMR are removed; and (3) Samples with missing data on the main variables are dropped. These procedures yield in a final sample of 7,419 firm-year observations from 1,006 unique firms. Additionally, we winsorize all continuous variables at the 1% and 99% quantiles to mitigate the influence of potential outliers.

## Identification strategy and model specification

We examine the effects of the NAMR on CSR by constructing a generalized DID model [6,10]. The NAMR has a direct and substantial influence on companies engaged in financialization since it is the first comprehensive and uniform regulatory framework for the asset management sector. Firms with a higher levels of asset financialization are expected to experience a more pronounced policy shock than those with a lower degree of financialization. Thus, we use a generalized DID model to investigate the impact of financial market regulation on non-financial firms' CSR, based on firms' pre-policy levels of financialization. We draw on the framework established by Qin et al. and Yu et al. [6,10,44,57] to specify the model as follows:

$$CSR_{i,t} = \beta_0 + \beta_1 Post * Prefin + \beta_i cv_{i,t} + \gamma_t + \mu_i + \varepsilon_{i,t} \tag{1}$$

where $i$ and $t$ index firm and year, respectively. *CSR* is the dependent variable, measured using Bloomberg's ESG composite score. *Post* is a year dummy variable equals to 1 for years 2018 and after, and otherwise 0. *Prefin* is a proxy for the average financialization of firm $i$ over the 3-period before the NAMR. $cv_{i,t}$ is a standard vector of controls known to influence CSR. $\gamma_t$ represents firm fixed effects, $\mu_i$ captures Post fixed effects, and $\varepsilon_{i,t}$ is the error term. They further control for unobserved heterogeneity that might impact CSR performance over time.

## Variables

**CSR.** The key dependent variable in this study is corporate social responsibility (CSR). CSR encompasses corporations' voluntary efforts to exceed legal and financial obligations, contributing to stakeholders through ethical practices, social initiatives, and sustainability efforts [58]. Yoon et al describe CSR as a broad, discretionary concept that includes social, ethical, and environmental activities beyond regulatory requirements [59]. However, measuring CSR has been a persistent challenge due to its subjective and qualitative nature. ESG emerged as a response to this challenge by standardizing and quantifying corporate responsibility efforts, making them comparable and transparent across firms and industries. As a result, ESG is not separate from CSR but rather an institutionalized mechanism that translates CSR commitments into measurable environmental, social, and governance indicators. Park et al. conduct a text-mining analysis and found that CSR and ESG are frequently discussed together, but their conceptual focus differs: CSR is qualitative and voluntary, focusing on corporate goodwill and ethical obligations. ESG is quantifiable and investment-driven, providing a framework that integrates corporate responsibility into financial and risk assessments [60]. In other words, ESG serves as a mechanism to operationalize CSR, ensuring that corporate responsibility efforts are systematically assessed and financially material to investors.

Given the strong correlation between CSR and ESG, ESG may serve as a proxy for CSR, offering more systematic coverage and improved data availability. We utilize Bloomberg's ESG composite score. The Bloomberg ESG measure minimizes measurement mistakes by offering a consistent and thorough evaluation of CSR behaviors. The ESG score assesses the quality and comprehensiveness of a firm's environmental and social governance disclosure and reporting activities and is based on several quantitative and policy-relevant indicators. Compared to other ESG ratings, the Bloomberg ESG composite score is more comprehensive and includes sub-scores for the E, S, and G dimensions. Therefore, the higher the ESG score, the greater a firm's engagement in CSR activities. Furthermore, the extensive usage of Bloomberg CSR rating scores in the body of current CSR research lends credence to the data's legitimacy and authenticity [61–64].

**Financial market regulation.** To observe the policy effects, we introduce a year dummy variable (*Post*) and a treatment effect proxy variable (*Prefin*). Post is a year dummy variable that takes the value of 1 when the observation period is 2018 and beyond; otherwise, it takes the value of 0. *Prefin*, which represents the treatment effect, measures the extent to which firms were exposed to the NAMR, capturing the average financialization level of firm *i* during the three years preceding the regulation's implementation. The coefficient on the interaction term *Post*Prefin* respects the differential effect of the NAMR, which is the main policy impact of interest in our analysis.

**Control variables.** Building on previous literature [65,66], we control for several variables known to influence CSR. These include firm-level financial characteristics, such as firm size, gearing ratio, cash flow ratio, operating income growth rate, and governance structure, such as the number of directors, the proportion of independent directors, the proportion of shares held by the top ten shareholders, the degree of equity checks and balances, and the proportion of shares held by institutional investors, in all regression estimates. Detailed variable definitions are shown in Table 1.

## Empirical results

### Descriptive statistics and discussions

Table 2 presents the descriptive statistics for the main variables. The mean CSR score is 33.405, with a standard deviation of 8.243, a maximum value of 71.180, and a minimum value of 17.644. These statistics suggest substantial variation in CSR performance among the firms in the sample. The mean value of *Prefin* is 0.044, with a median of 0.017 and a maximum of 0.815. The minimum value is 0, indicating significant heterogeneity in the degree of financialization across the sampled firms. The descriptive statistics for the control variables fall within reasonable ranges, suggesting the absence of outliers or extreme observations that could distort the analysis.

### Baseline regression results

Table 3 presents the results from our baseline regression model (1). To demonstrate the robustness of our estimates, we employ a stepwise regression approach, gradually incorporating control variables into the model (1). In column (1) of

**Table 1. Variables.**

| Variables | Definition |
|---|---|
| CSR | ESG score |
| Post | The value equals 1 if the year is 2018 or after 2018, and 0 otherwise |
| Prefin | The average degree of financialization of firm *i* in the three periods prior to the implementation of the NAMR |
| Size | The natural logarithms of total assets |
| Lev | Total liabilities/ total assets |
| ROA | Net profit/the average balance of total assets |
| Cashflow | Net cash flows from operating activities/ total assets |
| Growth | The increase rate of main business revenue |
| Board | The natural logarithm of the number of board members |
| Top10 | (Number of shares held by Top 10 shareholders/ Total number of shares)*100 |
| Balance | Shareholding of 2nd largest shareholder/ Shareholding of 1st largest shareholder |
| INST | (Total number of shares owned by institutional investors/ Total number of share capital)*100 |
| TMTPay | The natural logarithm of total executive compensation |
| Separate | Separation of ownership and operation |

Note: This table offers a comprehensive exposition of the study's variables, detailing their precise definitions and the methods employed for their measurement.

**Table 2. Descriptive statistical of variables.**

| Variables | N | Mean | Std. Dev. | Median | max | min |
|---|---|---|---|---|---|---|
| CSR | 7419 | 33.405 | 8.243 | 30.953 | 71.18 | 17.644 |
| Prefin | 7419 | 0.044 | 0.076 | 0.017 | 0.815 | 0 |
| Size | 7419 | 23.464 | 1.310 | 23.334 | 28.636 | 19.56 |
| Lev | 7419 | 0.476 | 0.194 | 0.487 | 1.698 | 0.008 |
| ROA | 7361 | 0.048 | 0.070 | 0.039 | 0.955 | -0.644 |
| Cashflow | 7419 | 0.063 | 0.070 | 0.058 | 0.664 | -0.463 |
| Growth | 7361 | 0.163 | 0.356 | 0.105 | 2.14 | -0.498 |
| Board | 7419 | 2.167 | 0.203 | 2.197 | 2.833 | 1.099 |
| Top10 | 7419 | 59.601 | 15.671 | 59.767 | 92.659 | 24.593 |
| Balance | 7419 | 0.344 | 0.285 | 0.249 | 1 | 0.002 |
| INST | 7419 | 55.496 | 21.649 | 57.964 | 94.24 | 4.605 |
| TMTPay | 7394 | 14.978 | 0.746 | 14.886 | 18.584 | 11.321 |
| Separate | 6900 | 5.475 | 8.004 | 0.005 | 29.983 | 0 |

Note: This table presents summary statistics for key variables, including CSR, Prefin, firm fundamentals (Size, Lev, ROA, Cashflow, Growth, Board), and ownership structure (Top10, Balance, INST, TMTPay, Separate). The data are sourced from the CSMAR database.

Table 3, we present the regression results without key control variables. The coefficient for *Post\*Prefin* is negative and statistically significant at the 1% level, providing initial support for hypothesis 1a.

Column (2) of Table 3 shows the regression result, which incorporates all the control variables. The coefficient for *Post\*Prefin* is -0.069, statistically significant at the 1% level, further supporting hypothesis 1a. These results indicate a significant negative impact of the NAMR on non-financial firms' CSR performance. In terms of economic significance, a 1 standard deviation increase in firms' degree of financialization before the enforcement of the NAMR results in a 0.059 percentage point reduction in CSR performance after the implementation of the policy (0.069 * 0.869 = 0.059961).

## Robustness tests

**Parallel trend test.** The parallel trend hypothesis is the fundamental premise of the DID methodology for policy evaluation. In our research context, this assumption demands that, prior to the NAMR, there is no systematic difference in the trend of changes in the distribution of CSR among listed firms with different degrees of financialization. Following Jacobson et al. [67], we apply an event study approach using Model (1), designating 2015 as the baseline year. Subsequently, we introduce interaction terms of dummy variables with the treatment intensity variable (*Prefin*) for each year, spanning two years before, the current year, and four years after the enforcement of the NAMR, denoted as pre_2, pre_1, current, post_1, post_2, and post_3, post_4, respectively.

Table 4 presents the regression results. The coefficients of pre_2 and pre_1 are not statistically significant, suggesting that before the implementation of the NAMR, the distribution of CSR among non-financial listed firms did not systematically differ based on their degree of financialization, which passed the parallel trend test. The coefficients for the year of policy implementation and the subsequent four years are notably negative. This implies that the NAMR has an impact on CSR, thereby providing preliminary confirmation for hypothesis 1a.

**Placebo test.** In an ideal scenario, the NAMR would serve as a strictly exogenous shock for non-financial firms. However, the reality is that various observable and unobservable factors inevitably influence the execution of any policy. To assess whether unaccounted factors might bias the results of Model (1), we undertake a placebo test utilizing random event simulation. Specifically, we randomly generate both the year of policy implementation (*Post*) and firms' financialization degree (*Prefin*). Subsequently, we conduct a regression incorporating the randomly generated interaction

**Table 3. Baseline regression results.**

| Variables | (1) | (2) |
|---|---|---|
| | CSR | CSR |
| Post*Prefin | -0.090*** | -0.069*** |
| | (-7.00) | (-5.52) |
| Size | | 0.337*** |
| | | (11.48) |
| Lev | | -0.087*** |
| | | (-4.93) |
| ROA | | 0.017 |
| | | (1.47) |
| Cashflow | | 0.011 |
| | | (1.27) |
| Growth | | -0.649 |
| | | (-1.63) |
| Board | | -0.014 |
| | | (-0.92) |
| Indep | | 0.023* |
| | | (1.78) |
| Top10 | | -0.039 |
| | | (-1.63) |
| Balance | | 0.052*** |
| | | (3.34) |
| INST | | 0.071*** |
| | | (2.73) |
| TMTPay | | 0.045*** |
| | | (3.02) |
| Separate | | 0.004** |
| | | (2.09) |
| Constant | -0.556*** | -0.435*** |
| | (-36.24) | (-20.32) |
| Observations | 7,419 | 6,820 |
| R-squared | 0.517 | 0.531 |
| FIRM FIXED EFFECT | Yes | Yes |
| Year Fixed Effect | Yes | Yes |

Note: This table reports the regression results on the effect of the NAMR on CSR. The dependent variable is CSR, with Post*Prefin as the key independent variable. Controls include firm fundamentals (Size, Lev, ROA, Cashflow, Growth, Board) and ownership structure (Indep, Top10, Balance, INST, TMTPay, Separate). Firm and year fixed effects are included. Robust standard errors are clustered at the firm level, with t-statistics in parentheses;

***$p < 0.01$,

**$p < 0.05$,

*$p < 0.1$. as in the table below.

term *Post*Prefin* into the model (1), repeating this process 1000 Posts. The results of the placebo test, depicted in Fig 1, illustrate that the regression coefficients of the core explanatory variables are normally distributed around zero in the random sample of 1000 iterations. This provides compelling evidence that the observed decrease in CSR investment is indeed attributable to implementing the NAMR and no other unaccounted factors.

**Table 4. Results of the parallel trend test.**

| Variables | (1) |
|---|---|
| | CSR |
| pre_2 | -0.008 |
| | (-0.36) |
| pre_1 | -0.022 |
| | (-0.95) |
| current | -0.052*** |
| | (-2.89) |
| post_1 | -0.040** |
| | (-2.13) |
| post_2 | -0.036* |
| | (-1.87) |
| post_3 | -0.051** |
| | (-2.25) |
| post_4 | -0.140*** |
| | (-3.22) |
| Size | 0.336*** |
| | (11.44) |
| Lev | -0.087*** |
| | (-4.92) |
| ROA | 0.017 |
| | (1.44) |
| Cashflow | 0.011 |
| | (1.27) |
| Growth | -0.661* |
| | (-1.66) |
| Board | -0.014 |
| | (-0.94) |
| Indep | 0.023* |
| | (1.77) |
| Top10 | -0.039 |
| | (-1.62) |
| Balance | 0.052*** |
| | (3.36) |
| INST | 0.071*** |
| | (2.74) |
| TMTPay | 0.045*** |
| | (3.01) |
| Separate | 0.004** |
| | (2.12) |
| Constant | -0.436*** |
| | (-20.34) |
| Observations | 6,820 |
| R-squared | 0.532 |
| FIRM FIXED EFFECT | Yes |
| Year Fixed Effect | Yes |

Note: This table reports the results of a parallel trend test using the event study approach. The dependent variable is CSR, and the key independent variables are represented by indicators for the pre-implementation, implementation, and post-implementation periods. Specifically, pre_2 and pre_1 denote the two years before the policy was implemented, current represents the year of implementation, and post_1 to post_4 correspond to the first to fourth years after the policy was enacted.

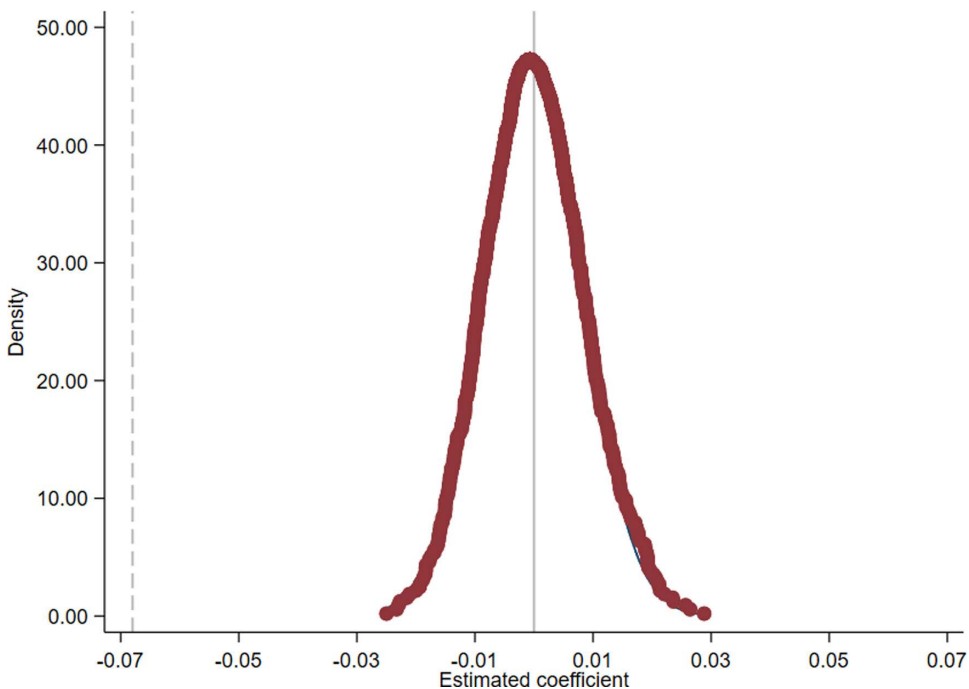

**Fig 1. Placebo test.** Note: Fig 1 shows the results of the placebo test. Each data point represents the estimated effect under different hypothetical scenarios, with the horizontal axis representing the effect estimates. Notably, the effect estimates for all data points remain close to zero, indicating that no significant changes are observed in the placebo scenario, similar to the main intervention effect.

**Exclusion of other policies.** To ensure that the observed effects are not confounded by other policies, such as monetary policy or U.S.-China trade frictions, we validate their exclusion with reference to Zhang et al.'s study [7]. First, Monetary Policy Impact: Broad money supply (M2) is used as a proxy for monetary policy. An interaction term between *M2* and *Post* is added to Model (1) to account for its potential influence. The results in column (1) of Table 5 indicate that monetary policy does not significantly impact the findings. Second.U.S.-China Trade Frictions: To address the potential interference of trade frictions, an interaction term between firms' export dependence (*ED*) and *Post* is included in Model (1). The results, shown in column (2) of Table 5, confirm that the effects of trade frictions between China and the U.S. are not driving the results. These analyses provide robust evidence that the observed changes in corporate investment are primarily attributable to the new regulations on capital management rather than other concurrent policy influences.

**Redefining the treatment group and PSM-DID.** Referring to Gao et al. [10], we reclassify firms based on the degree of financialization and create the treatment effect dummy variable "treat". Firms are assigned to the treatment group if their degree of financialization is higher than the median of *Prefin* (treat = 1). Otherwise, firms are assigned to the control group (treat = 0). We include the interaction term *Post*treat* instead of the interaction term *Post*Prefin* as the core explanatory variable in the regression model. The result is presented in column (1) of Table 7, showing a regression coefficient of -0.079, significant at the 1% level, supporting hypothesis 1a. Furthermore, we employ propensity score matching (PSM) to match samples based on firm characteristics to mitigate sample selection bias, subsequently re-conducting the DID analysis. Table 6 reports the Balance test results for PSM, indicating that after the propensity score matching treatment, the difference in covariate means between the treatment and control groups was not significant. The robustness of our findings is highlighted by the regression outcome, which is shown in column (2) of Table 7. It shows that the coefficient of *Post*treat* stays significantly negative at the 1% level.

**Table 5. Results of the exclusion of other policies.**

| Variables | (1) | (2) |
|---|---|---|
| | CSR | CSR |
| Post*Prefin | -0.069*** | -0.081*** |
| | (-5.52) | (-3.76) |
| Post*M2 | 0.000*** | |
| | (29.46) | |
| Post*ED | | 0.001 |
| | | (1.12) |
| Size | 0.337*** | 0.448*** |
| | (11.48) | (10.55) |
| Lev | -0.087*** | -0.114*** |
| | (-4.93) | (-4.56) |
| ROA | 0.017 | 0.003 |
| | (1.47) | (0.20) |
| Cashflow | 0.011 | 0.011 |
| | (1.27) | (0.88) |
| Growth | -0.649 | -0.598 |
| | (-1.63) | (-1.04) |
| Board | -0.014 | -0.029 |
| | (-0.92) | (-1.29) |
| Indep | 0.023* | 0.007 |
| | (1.78) | (0.40) |
| Top10 | -0.039 | -0.032 |
| | (-1.63) | (-0.93) |
| Balance | 0.052*** | 0.031 |
| | (3.34) | (1.47) |
| INST | 0.071*** | 0.102*** |
| | (2.73) | (2.87) |
| TMTPay | 0.045*** | 0.017 |
| | (3.02) | (0.83) |
| Separate | 0.004** | 0.001 |
| | (2.09) | (0.48) |
| Constant | -0.435*** | -0.382*** |
| | (-20.32) | (-12.61) |
| Observations | 6,820 | 3,829 |
| R-squared | 0.531 | 0.557 |
| FIRM FIXED EFFECT | Yes | Yes |
| Year Fixed Effect | Yes | Yes |

Note: This table presents results after excluding alternative policy influences. The dependent variable is CSR, with Post*Prefin as the key independent variable. In addition, Post*M2 and Post*ED are included as control variables to account for the effects of monetary policy and US-China trade frictions, respectively.

**Removing interference samples.** We excluded the major public health events that occurred during the sample period. Given the extensive outbreak of the COVID-19 epidemic at the end of 2019, such events significantly impact the social responsibility needs of China's listed firms, leading to changes in CSR decisions. To mitigate the crisis period's influence on results, the study excludes the 2020 sample. Column (3) of Table 7 shows the regression result, and the

**Table 6. Balance test results.**

| Variables | Unmatched/ Matched | Mean | | %bias | %reduce\|bias\| | t-test | |
|---|---|---|---|---|---|---|---|
| | | Treated | Control | | | t | p>\|t\| |
| Size | U | -0.055 | 0.046 | -10.4 | 82.1 | -4.37 | 0.000 |
| | M | -0.548 | -0.073 | 1.9 | | 0.77 | 0.444 |
| Lev | U | -0.073 | 0.076 | -15.3 | 96.4 | -6.40 | 0.000 |
| | M | -0.072 | -0.067 | -0.5 | | -0.22 | 0.825 |
| ROA | U | 0.004 | 0.022 | -2.0 | 7.3 | -0.83 | 0.406 |
| | M | 0.004 | -0.011 | 1.8 | | 0.74 | 0.459 |
| Cashflow | U | -0.077 | 0.073 | -16.0 | 88.4 | -6.68 | 0.000 |
| | M | -0.075 | -0.092 | 1.8 | | 0.76 | 0.450 |
| Growth | U | -0.017 | -0.018 | -3.8 | 60.4 | -1.61 | 0.108 |
| | M | -0.017 | -0.018 | 1.5 | | 0.76 | 0.450 |
| Board | U | -0.090 | 0.089 | -18.2 | 98.0 | -7.64 | 0.000 |
| | M | -0.090 | -0.086 | -0.4 | | -0.15 | 0.881 |
| Indep | U | 0.041 | -0.055 | 10.2 | 86.0 | 4.26 | 0.000 |
| | M | 0.041 | 0.054 | -1.4 | | -0.57 | 0.570 |
| Top10 | U | -0.071 | 0.102 | -17.9 | 79.5 | -7.49 | 0.000 |
| | M | -0.069 | -0.035 | -3.7 | | -1.51 | 0.132 |
| Balance | U | 0.541 | -0.110 | 16.6 | 89.8 | 6.94 | 0.000 |
| | M | 0.514 | -0.080 | -1.7 | | -0.68 | 0.497 |
| INST | U | -0.086 | 0.110 | -20.0 | 97.9 | -8.39 | 0.000 |
| | M | -0.084 | -0.080 | -0.4 | | -0.17 | 0.867 |
| TMTPay | U | 0.025 | -0.099 | 12.9 | 91.5 | 5.42 | 0.000 |
| | M | 0.024 | 0.013 | 1.1 | | 0.45 | 0.655 |
| Separate | U | 4.735 | 6.318 | -19.3 | 97.2 | -8.06 | 0.000 |
| | M | 4.740 | 4.694 | 0.5 | | 0.24 | 0.809 |

Note: This table presents the results of the PSM balance test. In line with the degree of financialization, kernel matching is performed on a 1:1 basis, and the differences in covariate means between the treatment and control groups are not statistically significant.

coefficient is significant at the 1% level, indicating that the NAMR has a negative effect on CSR performance, which reaffirms our research findings.

**Other robustness tests.** We conduct additional robustness tests as follows: First, there is a substitution of the dependent variable with another measure. With reference to Qian [68], we use the Huazheng ESG rating composite score to assess CSR. The regression result in column (4) of Table 7 confirms the benchmarking results. Second, we substitute the regression methodology. We employ the OLS model and a high-dimensional fixed effects model (adding industry fixed effects) to analyze, and the significant negative regression coefficients in columns (5) and (6) of Table 7 again prove the negative impact of the new asset management regulations on CSR.

## Further analysis

### Mediating mechanism test

The empirical results indicate a significant decline in non-financial firms' CSR performance following the enforcement of the NAMR. As outlined in the research hypotheses, resource constraints and performance pressures drive managers, guided by shareholder value maximization, to prioritize high-return investment ventures over CSR initiatives. The NAMR,

**Table 7. Results of robustness tests.**

| Variables | (1) | (2) | (3) | (4) | (5) | (6) |
|---|---|---|---|---|---|---|
| | CSR | CSR | CSR | CSR | CSR | CSR |
| Post*Prefin (Post*treat) | -0.079*** | -0.100*** | -0.074*** | -0.032* | -0.068*** | -0.035* |
| | (-3.53) | (-2.98) | (-5.58) | (-1.71) | (-5.01) | (-1.86) |
| Size | 0.341*** | 0.367*** | 0.345*** | 0.345*** | 0.367*** | 0.379*** |
| | (11.57) | (8.07) | (11.04) | (7.90) | (27.79) | (8.20) |
| Lev | -0.087*** | -0.070*** | -0.093*** | -0.154*** | -0.093*** | -0.156*** |
| | (-4.92) | (-2.61) | (-4.96) | (-5.87) | (-7.38) | (-5.87) |
| ROA | 0.017 | 0.008 | 0.019 | 0.029* | -0.001 | 0.033* |
| | (1.41) | (0.41) | (1.51) | (1.66) | (-0.06) | (1.84) |
| Cashflow | 0.012 | 0.025** | 0.008 | -0.023* | 0.049*** | -0.025* |
| | (1.35) | (1.96) | (0.92) | (-1.77) | (4.22) | (-1.94) |
| Growth | -0.670* | -0.759 | -0.609 | -2.265*** | -0.440 | -2.167*** |
| | (-1.68) | (-1.23) | (-1.43) | (-3.82) | (-0.74) | (-3.60) |
| Board | -0.014 | -0.053** | -0.015 | 0.008 | 0.026** | 0.015 |
| | (-0.96) | (-2.25) | (-0.91) | (0.36) | (2.33) | (0.66) |
| Indep | 0.023* | 0.009 | 0.020 | 0.070*** | 0.050*** | 0.070*** |
| | (1.81) | (0.45) | (1.43) | (3.66) | (4.50) | (3.66) |
| Top10 | -0.045* | -0.044 | -0.042 | -0.017 | 0.034** | -0.008 |
| | (-1.86) | (-1.21) | (-1.59) | (-0.47) | (2.43) | (-0.22) |
| Balance | 0.055*** | 0.055** | 0.055*** | 0.050** | 0.096*** | 0.044* |
| | (3.54) | (2.35) | (3.35) | (2.16) | (9.99) | (1.90) |
| INST | 0.071*** | 0.084** | 0.056** | -0.013 | 0.067*** | 0.000 |
| | (2.73) | (2.18) | (2.01) | (-0.33) | (4.63) | (0.01) |
| TMTPay | 0.045*** | 0.068*** | 0.042*** | 0.026 | 0.104*** | 0.026 |
| | (3.02) | (2.98) | (2.66) | (1.16) | (9.53) | (1.15) |
| Separate | 0.004** | 0.004 | 0.003 | 0.002 | -0.000 | 0.003 |
| | (2.04) | (1.52) | (1.58) | (0.64) | (-0.09) | (0.99) |
| Constant | -0.433*** | -0.410*** | -0.428*** | 0.005 | -0.376*** | -0.434 |
| | (-20.19) | (-12.48) | (-19.35) | (0.17) | (-13.19) | (-1.61) |
| Observations | 6,820 | 3,625 | 5,872 | 6,820 | 6,820 | 6,820 |
| R-squared | 0.530 | 0.535 | 0.565 | 0.050 | 0.417 | 0.065 |
| FIRM FIXED EFFECT | Yes | Yes | Yes | Yes | Yes | Yes |
| Year Fixed Effect | Yes | Yes | Yes | Yes | Yes | Yes |

Note: This table presents the results of several robustness tests. The dependent variable is CSR and the key independent variable is Post*Prefin (Post*treat). Columns (1) and (2) of Table 7 report the results obtained by redefining the treatment group and applying PSM-DID approach. Column (3) reports the regression results after excluding the 2020 sample, while column (4) presents the result when using the Huazheng ESG rating composite score as an alternative measure of CSR. We also use both an ordinary least squares (OLS) model and a high-dimensional fixed effects model (including industry fixed effects) in our analysis; significant negative regression coefficients are observed in columns (5) and (6) of Table 7.

by restricting practices such as multi-layer nesting and channel businesses and imposing liability ceilings on specific asset management products, has reduced firms' financial asset returns. This decline has created short-term performance pressures, leading firms to scale back their CSR investments. Thus, we posit that the NAMR influences CSR by reducing the returns on financial asset investments.

Baron and Kenny's stepwise regression method has been used in numerous research to investigate mediating mechanisms. We adopt this approach for the mechanism test in this study, with the regression results presented in Table 8.

In column (1) of Table 8, the coefficient of *Post\*Prefin* is significantly negative at the 10% level, indicating that implementing the NAMR significantly reduces the return on financial asset investments. In column (2), *FinReturn* and *Post\*Prefin* are significantly negative, suggesting that NAMR reduces CSR investment by decreasing the return on financial asset investments.

**Table 8. The results of mediating mechanism test.**

| Variables | (1) FinReturn | (2) CSR |
|---|---|---|
| FinReturn | | -0.100* |
| | | (-1.85) |
| Post*Prefin | -0.012* | -0.053*** |
| | (-1.80) | (-2.75) |
| Size | 0.066*** | 0.536*** |
| | (3.97) | (10.96) |
| Lev | -0.006 | -0.112*** |
| | (-0.60) | (-3.79) |
| ROA | 0.112*** | 0.023 |
| | (18.61) | (1.24) |
| Cashflow | -0.031*** | -0.005 |
| | (-7.17) | (-0.39) |
| Growth | -2.055*** | -1.367** |
| | (-9.71) | (-2.21) |
| Board | -0.007 | -0.048** |
| | (-0.95) | (-2.18) |
| Indep | 0.001 | -0.018 |
| | (0.11) | (-0.93) |
| Top10 | -0.048*** | -0.012 |
| | (-3.80) | (-0.31) |
| Balance | -0.001 | 0.023 |
| | (-0.10) | (1.00) |
| INST | 0.012 | 0.075* |
| | (0.88) | (1.89) |
| TMTPay | -0.024*** | 0.040* |
| | (-3.17) | (1.83) |
| Separate | -0.002** | -0.003 |
| | (-2.42) | (-1.02) |
| Constant | -0.049*** | -0.477*** |
| | (-4.71) | (-16.10) |
| Observations | 3,991 | 3,991 |
| R-squared | 0.141 | 0.531 |
| FIRM FIXED EFFECT | Yes | Yes |
| Year Fixed Effect | Yes | Yes |

Note: This table presents the results of the mediation mechanism test using Baron and Kenny's stepwise regression mediation method. The dependent variable is CSR, with Post*Prefin as the key independent variable and FinReturn as the mediating variable. FinReturn represents the profitability of investments in financial assets, measured as the sum of investment income, gains and losses on fair value adjustments and foreign exchange gains and losses, divided by operating income.

However, the validity of this approach has been questioned by international scholars [69] due to its potential to reduce statistical test efficiency and introduce bias in effect estimates [70]. To address these issues, we employ Bootstrap testing to test the mediation mechanism. Furthermore, we also utilize the Sobel test to quantitatively evaluate the proportion of the mediating effect to the total effect, enhancing the robustness of our mechanism test outcomes. We conducted 1000 random samples at a 95% confidence level. Table 9 presents the results of the mediating mechanism test. The Bootstrap regression results demonstrate that the return on financial asset investments significantly mediates the relationship between the NAMR and CSR. Specifically, the coefficient of the indirect effect of financial asset investment returns is significantly negative (index = -0.012, 95% bias-corrected confidence interval [-0.021, -0.003]), with the ratio of the indirect effect to the total effect at 0.135 (Sobel z = -0.012, p < 0.005).

### Impact of the NAMR on CSR dimensions (E, S, G)

To further investigate the effects of the NAMR on the distinct dimensions of CSR, we utilize Bloomberg's ESG scoring system to classify CSR into three dimensions: environment (E), society (S), and governance (G). These dimensions represent specific aspects of corporate responsibility: E reflects the balance between business operations and ecological sustainability, S represents the balance between individual and societal interests, and G captures the interplay between internal corporate governance and external social governance [71,72]. Given that each dimension involves different implementation costs, firms may respond differently to financial market regulation. To analyze this, we replace the core variable in Model (1) with measures for E, S, and G to examine the NAMR's heterogeneous effects on these dimensions. Table 10 presents the sub-dimensional regression results.

In columns (1) and (3) of Table 10, the coefficients of the *Post\*Prefin* are significantly negative at the 1% level. In contrast, the coefficient in column (2) is insignificant, suggesting that the NAMR significantly diminishes firms' E-dimension and G-dimension CSR, with no significant negative impact on the S-dimension CSR. Notably, the coefficient of *Post\*Prefin* on firms' E in column (1) is -0.096, and that of *Post\*Prefin* on firms' G in column (3) is -0.046, suggesting that the NAMR more profoundly impacts firms' E-dimension CSR.

## Discussion

Previous research on firm governance suggests that financial markets can serve as platforms for regulating corporate behavior [21]. Strengthening financial market regulation is essential not only for preventing and mitigating financial risks but also for guiding firms toward high-quality development. Using China's financial market as a case study, this paper examines the impact of financial market regulation on CSR behavior. We propose that stricter financial market regulation

**Table 9. The results of Bootstrap Method and Sobel Test.**

| Mediating Variable: FinReturn | | | | | | |
|---|---|---|---|---|---|---|
| Bootstrap Method | | Coefficient | Std. Err. | P>z | 95% confidence interval | |
| | | | | | LLCI | ULCI |
| | Indirect_effect_ | -0.012 | 0.005 | 0.011 | -0.021 | -0.003 |
| | Direct_effect | -0.077 | 0.016 | 0.000 | -0.110 | -0.045 |
| Sobel Test | | Coefficient | Std. Err. | P>z | | |
| | Indirect_effect | -0.012 | 0.005 | 0.017 | | |
| | Direct_effect | -0.077 | 0.018 | 0.000 | | |
| | Total_effect | -0.090 | 0.017 | 0.000 | | |
| | Proportion of total effect that is mediated | 0.135 | | | | |

Note: This table presents the results from the Bootstrap Method and the Sobel Test. In this analysis, FinReturn serves as the mediating variable, and "Indirect_effect" denotes the mediating effect of FinReturn on the relationship between Post\*Prefin and CSR.

**Table 10. Impact of the NAMR on CSR dimensions (E, S, G).**

| Variables | (1) | (2) | (3) |
|---|---|---|---|
| | E | S | G |
| Post*Prefin | -0.096*** | -0.000 | -0.046*** |
| | (-6.53) | (-0.01) | (-3.28) |
| Size | 0.311*** | 0.393*** | 0.092*** |
| | (9.01) | (12.12) | (2.80) |
| Lev | -0.065*** | -0.074*** | -0.091*** |
| | (-3.14) | (-3.83) | (-4.66) |
| ROA | 0.029** | 0.014 | -0.009 |
| | (2.11) | (1.07) | (-0.69) |
| Cashflow | 0.009 | 0.018* | -0.001 |
| | (0.88) | (1.87) | (-0.08) |
| Growth | -0.818* | -0.790* | 0.345 |
| | (-1.74) | (-1.79) | (0.78) |
| Board | -0.006 | -0.040** | 0.014 |
| | (-0.34) | (-2.44) | (0.81) |
| Indep | 0.021 | 0.013 | 0.023 |
| | (1.43) | (0.95) | (1.58) |
| Top10 | -0.061** | -0.027 | 0.026 |
| | (-2.15) | (-1.01) | (0.98) |
| Balance | 0.050*** | 0.036** | 0.037** |
| | (2.76) | (2.12) | (2.14) |
| INST | 0.110*** | 0.066** | -0.060** |
| | (3.61) | (2.32) | (-2.07) |
| TMTPay | 0.041** | 0.029* | 0.041** |
| | (2.37) | (1.74) | (2.48) |
| Separate | 0.003 | 0.003* | 0.002 |
| | (1.64) | (1.69) | (1.20) |
| Constant | -0.388*** | -0.317*** | -0.396*** |
| | (-15.41) | (-13.40) | (-16.57) |
| Observations | 6,820 | 6,820 | 6,820 |
| R-squared | 0.402 | 0.374 | 0.381 |
| FIRM FIXED EFFECT | Yes | Yes | Yes |
| Year Fixed Effect | Yes | Yes | Yes |

Note: This table presents the regression results examining the impact of NAMR on environmental (E), social (S), and governance (G). The dependent variables are E, S, and G, with Post*Prefin serving as the key independent variable.

may exert either a positive or negative influence on CSR. To test this hypothesis, we leverage China's stringent regulatory policy in the capital management industry, constructing a quasi-natural experiment and employing a generalized DID model.

Analyzing a dataset of non-financial listed firms from 2015 to 2022, our findings reveal that tighter financial market regulation leads to poorer CSR performance among non-financial firms, supporting hypothesis 1a. Specifically, the implementation of the NAMR influences firms' investment decisions, consistent with Bojiang's findings. Further analysis of the mechanism reveals that the NAMR impacts CSR by reducing returns on financial asset investments. This aligns with the

shareholder value maximization view, which suggests that firms prioritize profitability over sustainability when financial resources are constrained. This finding highlights a broader theoretical tension between shareholder primacy and stakeholder theory. While financial market regulation curbs speculative behavior [54], it may inadvertently exacerbate trade-offs between profitability and socially responsible practices. These results underscore the need for integrated regulatory frameworks that balance economic and social objectives.

Additionally, our study examines the NAMR's effects on different dimensions of CSR—environment (E), society (S), and governance (G). The results show that the NAMR significantly affects the E and G dimensions of CSR, with no notable impact on the S dimension. Among these, the environmental dimension is the most severely affected. This disparity may stem from the fact that Chinese firms initially allocate fewer resources to the social dimension compared to environmental and governance aspects, leaving the S dimension less sensitive to regulatory changes. Unlike environmental initiatives (which require immediate investment) and governance compliance (which is highly regulated), social responsibility investments often have longer adjustment periods. The NAMR may trigger changes that manifest beyond the observed time frame. Moreover, under China's regulatory system, the government has strengthened environmental policies in recent years, prompting firms to invest in green innovation and environmental governance to reduce pollutant emissions. In the capital market, governance information tends to hold higher value for investors than social performance, driving firms to prioritize investments in the E and G dimensions. Furthermore, the externalities of environmental and social responsibilities cannot be fully internalized, creating a conflict between improving environmental and social performance and maximizing corporate profits [73]. Consequently, as the NAMR reduces firms' profits, their ability to sustain environmental performance is disproportionately hindered. These findings highlight the complex interplay between financial regulation, corporate behavior, and the competing priorities of profitability and sustainability.

## Conclusion

This study examines the impact of China's stringent asset management regulation on corporate social responsibility (CSR), using data from non-financial listed firms from 2015 to 2022. The findings show that tighter financial market regulation significantly reduces CSR performance. Mechanism tests suggest that this effect is driven by reduced returns on financial asset investments. Additionally, its effects vary across dimensions, with the most pronounced impacts observed in environmental (E) and governance (G) practices, while social (S) initiatives remain largely unaffected.

By leveraging China's financial market as a case study, this research provides valuable insights into how regulatory environments shape corporate behavior. It expands the understanding of CSR determinants beyond traditional drivers like firm characteristics and managerial decisions, demonstrating how external shocks influence corporate priorities. Moreover, the study challenges the conventional view that stricter regulation universally improves firm behavior, instead revealing the trade-offs between regulatory compliance and broader societal objectives.

The findings carry significant implications for policymakers and firms. The decline in environmental and governance-related initiatives suggests that stringent financial regulations may unintentionally weaken firms' commitment to social responsibility. To counteract these effects, policymakers could introduce complementary measures such as tax incentives, subsidies for green initiatives, or improved access to sustainable financing. Additionally, promoting awareness of the long-term strategic value of CSR can help firms embed social responsibility into their core operations, even amid financial constraints.

For firms, it is crucial to align social investments with operational and financial objectives to maximize their impact under limited resources. Managers should also consider innovative financing tools—such as green bonds or sustainability-linked loans, to sustain responsible practices while complying with regulatory requirements.

This study has several limitations that open avenues for further exploration. First, as the focus is on China, the findings are most applicable to developing economies, and their relevance to developed markets with advanced financial systems

may be limited. Future studies could investigate how financial regulations influence corporate behavior in such contexts, where institutional frameworks differ significantly.

Second, the relatively short observation period captures only the immediate effects of the regulation, limiting the ability to assess its medium- and long-term impacts. Extending the timeframe could provide a clearer understanding of whether financial regulations have lasting effects on corporate behavior and the broader economic outcomes.

Third, the study does not account for regional and industry-specific heterogeneity, which may influence how firms respond to regulation. Differences in financial market conditions and institutional environments across regions and sectors may result in heterogeneous effects. Future research could incorporate contextual factors such as industry characteristics or regional differences to better understand these dynamics. Additionally, firm-specific resources and strategies likely play a critical role in shaping responses to regulatory changes, suggesting another promising avenue for exploration.

By addressing these gaps, future research can deepen the understanding of the interplay between financial regulation and corporate responsibility, offering actionable insights for both policymakers and practitioners.

## Author contributions

**Conceptualization:** Le Zhu, Quan Zhang.

**Data curation:** Le Zhu, Quan Zhang.

**Formal analysis:** Le Zhu.

**Funding acquisition:** Le Zhu.

**Investigation:** Le Zhu.

**Methodology:** Le Zhu.

**Project administration:** Yichuan Wang.

**Resources:** Yichuan Wang.

**Supervision:** Yichuan Wang, Quan Zhang.

**Writing – review & editing:** Yichuan Wang, Quan Zhang.

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
