## [Decision Letter · Decision Letter 0]

22 Dec 2024

PONE-D-24-48995

Financial Market Regulation and Corporate Social Responsibility: Evidence from China’s New Asset Management Regulation

PLOS ONE

Dear Dr. Wang,

Thank you for submitting your manuscript to PLOS ONE. After careful consideration, we feel that it has merit but does not fully meet PLOS ONE’s publication criteria as it currently stands. Therefore, we invite you to submit a revised version of the manuscript that addresses the points raised during the review process.

We look forward to receiving your revised manuscript.

Kind regards,

Saddam A. Hazaea, Postdoctoral

Academic Editor

PLOS ONE

Journal Requirements:

“This work was supported by the Key Program of National Social Science Foundation of China under Grant [No. 20AGL010].”

4. Thank you for uploading your study's underlying data set. Unfortunately, the repository you have noted in your Data Availability statement does not qualify as an acceptable data repository according to PLOS's standards.

Additional Editor Comments:

Dear authors

I hope you are doing well

Please address all the comments raised by reviewers. In addition, your paper needs to be restructured and requires full proofreading. Please note that some reviews ask you to cite specific articles, however, you can cite them only if you feel it's appropriate and will enhance your discussion. Otherwise, it's not compulsory.

Thank you

Reviewers' comments:

Reviewer's Responses to Questions

**Comments to the Author**

1. Is the manuscript technically sound, and do the data support the conclusions?

Reviewer #1: Partly

Reviewer #2: Partly

Reviewer #3: Yes

Reviewer #4: Yes

Reviewer #5: Partly

2. Has the statistical analysis been performed appropriately and rigorously? 

Reviewer #1: No

Reviewer #2: No

Reviewer #3: Yes

Reviewer #4: Yes

Reviewer #5: N/A

3. Have the authors made all data underlying the findings in their manuscript fully available?

Reviewer #1: Yes

Reviewer #2: No

Reviewer #3: Yes

Reviewer #4: Yes

Reviewer #5: Yes

4. Is the manuscript presented in an intelligible fashion and written in standard English?

Reviewer #1: Yes

Reviewer #2: No

Reviewer #3: Yes

Reviewer #4: Yes

Reviewer #5: Yes

5. Review Comments to the Author

Reviewer #1: Research Gaps:

The research gaps in the paper are not clearly defined. It would be helpful to elaborate on the specific areas where further research is needed. Providing a more detailed context will strengthen the paper's contribution to the field.

Citations:

There is a need for more citations to support the arguments presented. Specifically, consider integrating recent studies to demonstrate the relevance and novelty of your research.

Citation Style Errors:

The citations are inconsistent with the required styles (APA 7 & IEEE). Below are the specific issues:

APA 7th edition requires that author names be listed with the year in parentheses, for example, "Qin et al. (2023)".

Some citations are missing publication details or have incorrect formatting. For instance, "Yu et al. (2024)" and "Drucker, 2012" should be corrected.

Citation for Gao et al. (2023) is incomplete, and "Baron and Kenny's (1986)" should be fully cited with complete details.

Please ensure consistency in the citation style throughout the manuscript.

Comments on the Use of the Difference-in-Differences (DID) Model:

Appropriateness of the DID Model:

The Difference-in-Differences (DID) model is an excellent choice when you are evaluating the causal impact of a policy intervention or treatment that affects a treatment group, while a control group is unaffected. It helps account for time trends that might affect both groups in the absence of the intervention, essentially comparing the differences in outcomes before and after the treatment for both groups.

Contextual Fit:

For the DID model to be appropriate, it is critical that the study satisfies the parallel trends assumption. This assumption posits that, in the absence of the treatment, the treatment and control groups would have followed similar trends over time. You should explicitly justify this assumption and, if possible, provide visual or statistical evidence (e.g., pre-treatment trends) to support that the treatment and control groups were on similar trajectories prior to the intervention.

Data Requirements:

The DID model requires data that covers pre-treatment and post-treatment periods for both the treatment and control groups. Please ensure that you have sufficient observations in both periods for both groups to draw reliable conclusions. Additionally, the model assumes that no other events or interventions during the study period are affecting the outcome, apart from the treatment being studied.

Possible Confounding Factors:

One potential issue with the DID model is the presence of confounding variables that might impact the treatment and control groups differently over time. It's crucial to control for variables that might influence the outcome, particularly if they vary across time or groups. You may want to include relevant covariates in your model or perform robustness checks to address any potential confounding.

Interpretation of Results:

The DID model provides an estimate of the average treatment effect for the treated units (ATT). Ensure that the interpretation of the coefficients is clear and that you’re explaining the economic or policy implications of the results. It's also important to clarify whether your results reflect the short-term or long-term impact of the intervention.

Suggestions for Improvement:

If you are not already doing so, consider performing placebo tests or sensitivity analyses to ensure that the DID results are robust. You could also examine whether the parallel trends assumption holds by plotting the outcome variable for both groups before the treatment.

The manuscript is generally well-structured, but some sections lack clarity, especially in the Literature Review and Methodology sections. I suggest revising these sections for better flow and coherence. In particular, the explanation of the [specific model/approach] is somewhat confusing, and it would be helpful to simplify the description for readers unfamiliar with the technique.

Additionally, some figures and tables need clearer labels and better descriptions in the captions.

Implications:

The manuscript currently lacks a discussion on the implications of the findings. It is crucial to include the following:

Practical Implications: How do the findings apply to real-world scenarios? What recommendations can be made for practitioners or policymakers?

Theoretical Implications: How do the results contribute to advancing theory in the field? What new insights does the paper provide?

Social Implications: How does the research impact society? Are there any ethical, cultural, or social considerations arising from the findings?

Reviewer #2: This paper employs the NAMR in China as a quasi-natural experiment to investigate the impact of financial regulation on firms' CSR investment. The authors find a decline in firms' CSR performance after experiencing the more stringent financial market regulation. They claim the channel of this causal effect is through the reduction of financial asset return. Although this is an interesting topic, the authors need many revisions before submitting it to a journal.

Major Comments:

1. Not enough information about the NAMR event. The authors fail to have a comprehensive introduction of the exogenous shock they employed. They should discuss the event in detail, including the precise time that the new policy was enacted, how the financial investment regulations are strengthened (in detail!), etc. More importantly, I am curious whether the policy was enacted nationwide concurrently or whether the authority started to have a trial in some provinces and extended it to other regions later. It would be vital to the DID setting in the empirical analysis.

2. The selection of the treatment group is questionable. The authors use the level of financialization of non-financial firms before the event as a cutoff to separate the sample. It could be endogenous since firms with different levels of financialization are likely to have different preferences (motivation) for CSR. Thus, this setting may not capture the causal effect. I will suggest the authors look for cleaner treatment identification.

3. The channels of the effect are not solid. Although the authors claim the reduction of financial investment profits, they do not illustrate the difference in financial profits of firms before and after the event. Additionally, the authors separate the hypothesis into two parts, the value of shareholders and stakeholders, where they assume the shareholders would advocate reducing the CSR expense after strengthening the regulation, while stakeholders would be the opposite. Do the results indicate that firms only care about the shareholders' interests while abandoning those for stakeholders? I will suggest the authors to more carefully diagnose their channels to explain this effect.

Minor comments:

4. The table format needs to be more professional. There is no caption to explain the regression econometrics of each table, and some inputs are not in the same line.

5. Paper writing needs to be polished.

Reviewer #3: Using China’s New Asset Management Regulation as an exogenous shock event, this study employs a generalized DID model to analyse and test how firms perceive or perform their social responsibility under strict financial market regulation. It is an interesting work, The following issues need to be resolved before the article is published:

Comment 1�It may be necessary to highlight the research significance of the paper in the abstract.

Comment 2

The Introduction exhibits a commendable structure and organization. However, it's pivotal to accentuate the novelty and unique contributions of this paper within the existing body of literature. This can be effectively showcased by delineating the primary contributions to the related works. Consider citing the most recent relevant literature to enrich the introduction.

Comment 3: Please give more detailed descriptive stats.

Comment 4:Occasional grammar errors persist throughout the manuscript, notably concerning missing articles such as "the," "a," and "an." A thorough spellcheck to rectify these minor issues is advised. Additionally, some sentences tend to be overly long, impacting readability. Consider breaking these lengthy sentences into shorter, more digestible ones for enhanced readability.

Reviewer #4: This article examines the impact of China's New Asset Management Regulation (NAMR) on corporate social responsibility (CSR). The research methodology is sound and holds both practical and theoretical value. However, there are several issues within the article that warrant discussion.

1.The article exhibits a conflation of concepts, equating Environmental, Social, and Governance (ESG) with corporate social responsibility (CSR), which occurs to varying degrees in both the theoretical and empirical analysis sections. In reality, while CSR and ESG are related, they are distinct. CSR refers to the social responsibilities that a company assumes in its business activities, whereas ESG encompasses the company's performance in environmental, social, and governance aspects.

2.In the appendix, the title of Figure 1 is incorrect.

3.The specific results of the Propensity Score Matching (PSM) are not presented, making it impossible to assess the appropriateness of the PSM method used.

4.Due to the author's conflation of ESG and CSR concepts, it is redundant to separately discuss the different dimensions of CSR, namely the E, S, and G aspects. It is recommended that these sections be removed.

5.The article overlooks the impact of other financial market policies in China when analyzing the effects of NAMR on corporate CSR, casting doubt on the reliability of the article's conclusions.

Reviewer #5: Strengths:

Addresses a significant gap in literature by exploring the relationship between financial market regulation and corporate social responsibility (CSR).

Employs a robust methodological approach using a quasi-natural experiment and generalized difference-in-differences model.

Provides nuanced insights through analysis of different CSR dimensions (environmental, social, and governance).

Enhances understanding through mediating mechanism test.

Areas for improvement:

Literature review could be more concise and focused on key arguments relevant to hypotheses.

Discussion section requires stronger linkages between findings and broader theoretical implications.

Minor grammatical and stylistic issues need addressing to enhance overall clarity.

More comprehensive discussion of study limitations and future research directions is warranted.

6. PLOS authors have the option to publish the peer review history of their article (what does this mean? ). If published, this will include your full peer review and any attached files.

**Do you want your identity to be public for this peer review?** For information about this choice, including consent withdrawal, please see our Privacy Policy .

Reviewer #1: **Yes: ** Mohammad Rakibul Islam Bhuiyan

Reviewer #2: No

Reviewer #3: No

Reviewer #4: No

Reviewer #5: **Yes: ** Jeenchen Chen, ChFC®, CLU®, EA

---

## [Author Response · Author response to Decision Letter 1]

3 Feb 2025

Reviewer #1

Comment 1: The research gaps in the paper are not clearly defined. It would be helpful to elaborate on the specific areas where further research is needed. Providing a more detailed context will strengthen the paper's contribution to the field. There is a need for more citations to support the arguments presented. Specifically, consider integrating recent studies to demonstrate the relevance and novelty of your research.

Response: We sincerely appreciate your suggestion to emphasize the research gaps. Highlighting the unique contribution of our paper is indeed critical to its positioning within the academic literature. To address this, we have revised the introduction section to provide a more comprehensive research background related to financial market regulation and corporate social responsibility. Additionally, we have incorporated and cited recent, highly relevant literature to strengthen our contribution and clarify the unique aspects of our study. These references underscore the significance of our work in the context of evolving financial regulations and their impact on corporate practices. The newly added citations include:

1. Dong Y, Dong M, Tan S, Ge R. Shadow banking, financial regulation, and bank risk in China. Financ Res Lett. 2024;63: 105293. doi: https://doi.org/10.1016/j.frl.2024.105293

2. Jungo J, Madaleno M, Botelho A. The Effect of Financial Inclusion and Competitiveness on Financial Stability: Why Financial Regulation Matters in Developing Countries? Journal of Risk and Financial Management. 2022;15. doi:10.3390/jrfm15030122

3. Ullah W, Zubir ASM, Ariff AM. Exploring the moderating effect of regulatory quality on the relationship between financial development and economic growth/economic volatility for developed and developing countries. Borsa Istanbul Review. 2024;24: 934–944. doi: https://doi.org/10.1016/j.bir.2024.04.015

4. Delle Foglie A, Boukrami E, Vento G, Panetta IC. The regulators' dilemma and the global banking regulation: the case of the dual financial systems. Journal of Banking Regulation. 2023;24. doi:10.1057/s41261-022-00196-2

5. Ofoeda I, Amoah L, Anarfo EB, Abor JY. Financial inclusion and economic growth: What roles do institutions and financial regulation play? International Journal of Finance and Economics. 2024;29. doi:10.1002/ijfe.2709

6. Liu Q, Wu J. Strong financial regulation and corporate risk-taking: Evidence from a natural experiment in China. Financ Res Lett. 2023;54. doi:10.1016/j.frl.2023.103747

7. Zhang B, Guo M. Strong financial regulation and the intelligent transformation of enterprises. Econ Anal Policy. 2025;85: 186–207. doi: https://doi.org/10.1016/j.eap.2024.11.017

8. Jiang M, Zhou W, Song Y. Research on shadow banking, the new regulations on asset management and corporate finance. Stud of Int Financ. 2020;12: 63–72.

9. Liu C. Disruption of corporate financialization and labor cost growth: Evidence from China's new asset management rules. PLOS One. 2023;18. doi:10.1371/journal.pone.0286683

Comment 2: Citation Style Errors. The citations are inconsistent with the required styles (APA 7 & IEEE). Below are the specific issues: APA 7th edition requires that author names be listed with the year in parentheses, for example, "Qin et al. (2023)".

Some citations are missing publication details or have incorrect formatting. For instance, "Yu et al. (2024)" and "Drucker, 2012" should be corrected.

Citation for Gao et al. (2023) is incomplete, and "Baron and Kenny's (1986)" should be fully cited with complete details.

Please ensure consistency in the citation style throughout the manuscript.

Response: Thank you for pointing out the inconsistencies in the citation style. We sincerely apologize for the oversight and have carefully revised all citations in the manuscript to ensure consistency and compliance with the required PLOS ONE citation format. We have also thoroughly reviewed the entire manuscript and have verified all references for accuracy and completeness.

Comment 3: Comments on the Use of the Difference-in-Differences (DID) Model:

Appropriateness of the DID Model:

The Difference-in-Differences (DID) model is an excellent choice when you are evaluating the causal impact of a policy intervention or treatment that affects a treatment group, while a control group is unaffected. It helps account for time trends that might affect both groups in the absence of the intervention, essentially comparing the differences in outcomes before and after the treatment for both groups.

Contextual Fit:

For the DID model to be appropriate, it is critical that the study satisfies the parallel trends assumption. This assumption posits that, in the absence of the treatment, the treatment and control groups would have followed similar trends over time. You should explicitly justify this assumption and, if possible, provide visual or statistical evidence (e.g., pre-treatment trends) to support that the treatment and control groups were on similar trajectories prior to the intervention.

Data Requirements:

The DID model requires data that covers pre-treatment and post-treatment periods for both the treatment and control groups. Please ensure that you have sufficient observations in both periods for both groups to draw reliable conclusions. Additionally, the model assumes that no other events or interventions during the study period are affecting the outcome, apart from the treatment being studied.

Possible Confounding Factors:

One potential issue with the DID model is the presence of confounding variables that might impact the treatment and control groups differently over time. It's crucial to control for variables that might influence the outcome, particularly if they vary across time or groups. You may want to include relevant covariates in your model or perform robustness checks to address any potential confounding.

Interpretation of Results:

The DID model provides an estimate of the average treatment effect for the treated units (ATT). Ensure that the interpretation of the coefficients is clear and that you're explaining the economic or policy implications of the results. It's also important to clarify whether your results reflect the short-term or long-term impact of the intervention.

Suggestions for Improvement:

If you are not already doing so, consider performing placebo tests or sensitivity analyses to ensure that the DID results are robust. You could also examine whether the parallel trends assumption holds by plotting the outcome variable for both groups before the treatment.

Response: Thank you for your insightful comments regarding the use of the Difference-in-Differences (DID) model in our study. We greatly appreciate your suggestions, as they have helped us enhance the rigor and clarity of our analysis.

We agree that satisfying the parallel trends assumption is critical for the validity of the DID model. To address this, we have conducted a parallel trends test and included statistical results to demonstrate that the treatment and control groups followed similar trajectories prior to the intervention. These results are presented in the revised manuscript.

As per your suggestion, we have performed a placebo test to ensure the robustness of our findings. The placebo test results confirm that there are no significant differences in the outcome variable during periods when the treatment should not have had any effect, further validating the causal interpretation of our results. These findings are also included in the revised text.

We appreciate your valuable feedback and trust that these revisions address your concerns effectively. Please let us know if there are any additional areas where further clarification or improvement is needed.

Comment 4: The manuscript is generally well-structured, but some sections lack clarity, especially in the Literature Review and Methodology sections. I suggest revising these sections for better flow and coherence. In particular, the explanation of the [specific model/approach] is somewhat confusing, and it would be helpful to simplify the description for readers unfamiliar with the technique.

Additionally, some figures and tables need clearer labels and better descriptions in the captions.

Response: Thank you for your valuable feedback regarding the structure and clarity of the manuscript. We appreciate your suggestions and have taken steps to improve the flow, coherence, and readability of the Literature Review and Methodology sections, as well as the presentation of figures and tables. We have revised the Literature Review section to enhance its logical flow and ensure that the key themes of relevant studies are clearly presented. To address the confusion regarding the explanation of the [specific model/approach], we have simplified the description, ensuring it is accessible to readers unfamiliar with the technique. We have reviewed all figures and tables in the manuscript and made some improvements.

Comment 5: Implications: The manuscript currently lacks a discussion on the implications of the findings. It is crucial to include the following:

Practical Implications: How do the findings apply to real-world scenarios? What recommendations can be made for practitioners or policymakers?

Theoretical Implications: How do the results contribute to advancing theory in the field? What new insights does the paper provide?

Social Implications: How does the research impact society? Are there any ethical, cultural, or social considerations arising from the findings?

Response: Thank you for your suggestion. The significance of the research is a crucial aspect of our paper. In the Introduction, we emphasize the theoretical importance of our study, while in the conclusion, we reassert both its theoretical and practical significance. Specifically, we highlight how our research contributes to the development of CSR theory and offers practical insights for firms and policymakers.

Reviewer #2

Comment 1: Not enough information about the NAMR event. The authors fail to have a comprehensive introduction of the exogenous shock they employed. They should discuss the event in detail, including the precise time that the new policy was enacted, how the financial investment regulations are strengthened (in detail!), etc. More importantly, I am curious whether the policy was enacted nationwide concurrently or whether the authority started to have a trial in some provinces and extended it to other regions later. It would be vital to the DID setting in the empirical analysis.

Response: Thank you for your valuable suggestion. We recognize the importance of providing a comprehensive introduction to the NAMR event, and we have revised the manuscript to address this concern. We have added a dedicated section on the institutional background, which includes a detailed description of the NAMR event, including the exact timing of the policy enactment, the specific financial investment regulations that were strengthened, and the broader policy effects. To clarify, the policy was enacted nationwide at the same time, and there was no phased implementation or trial period in specific provinces before its nationwide rollout. We hope these additions clarify the policy context and strengthen the understanding of the empirical analysis. The revised section can be found in the end of introduction in the manuscript.

Comment 2�The selection of the treatment group is questionable. The authors use the level of financialization of non-financial firms before the event as a cutoff to separate the sample. It could be endogenous since firms with different levels of financialization are likely to have different preferences (motivation) for CSR. Thus, this setting may not capture the causal effect. I will suggest the authors look for cleaner treatment identification.

Response: We sincerely appreciate your advice for raising this important concern regarding the selection of the treatment group and the potential endogeneity issues associated with using the level of financialization of non-financial firms as the cutoff. Your insightful comment has allowed us to critically reassess and further strengthen our identification strategy.

We chose the level of financialization as the identifying variable for the following reasons:

Relevance to Policy Impact: The New Asset Management Regulation (NAMR) specifically targets the asset management activities of financial institutions, with a focus on curbing the "de-realization" of funds. This regulation has a direct impact on the financialization behaviors of firms, particularly those relying on financial assets or arbitrage. By using the pre-NAMR financialization level as the cutoff, we aim to capture the heterogeneous effects of the policy across firms with different financialization intensities, thereby providing a clearer identification of the NAMR's impact on firm behavior.

Alignment with Generalized DID Framework:

Second, the generalized DID model requires explicit treatment groups (firms significantly affected by the policy) and control groups (firms less affected by the policy). Firms with higher financialization levels are more likely to have relied on financial asset investments rather than production and business activities before the NAMR, making them more significantly affected by the regulation. Conversely, firms with lower financialization levels are less directly impacted. This distinction ensures that the treatment and control groups reflect the varying degrees of policy exposure.

Support from Prior Literature:

Several studies have successfully used financialization levels to define treatment and control groups in quasi-natural experimental settings, underscoring the feasibility of this approach. Representative studies include:

Xuan, S., Song, D., & You, G. (2024). Financial risk prevention and corporate green innovation: A quasi-natural experiment based on the new asset management regulation. Pacific-Basin Finance Journal, 88, 102566.

Li, M., & Huang, Y. (2024). Financial regulation and financial market stability: Evidence from stock price crash risk. Finance Research Letters, 69, 106196.

Jiang, T., & Wu, G. (2025). Can strict financial regulation improve analysts' forecast accuracy? Evidence based on a quasi-natural experiment in China. Finance Research Letters, 106752.

We acknowledge, however, the potential endogeneity concern raised by the reviewer. Firms with varying levels of financialization may indeed exhibit different preferences or motivations for corporate social responsibility (CSR), which could confound our analysis. To address this concern and further enhance the robustness of our findings, we have implemented the following adjustments:

Propensity Score Matching (PSM): We have redefined the treatment and control groups using a propensity score matching (PSM) approach. This method accounts for pre-existing differences in CSR preferences and ensures a more exogenous selection of the sample, minimizing bias.

Robustness Tests: To validate the causal inferences, we conducted additional robustness checks, including parallel trend tests, placebo tests, and variable substitution. The results consistently support our conclusions, demonstrating that our findings remain robust across different specifications.

Comment 3� The channels of the effect are not solid. Although the authors claim the reduction of financial investment profits, they do not illustrate the difference in financial profits of firms before and after the event. Additionally, the authors separate the hypothesis into two parts, the value of shareholders and stakeholders, where they assume the shareholders would advocate reducing the CSR expense after strengthening the regulation, while stakeholders would be the opposite. Do the results indicate that firms only care about the shareholders' interests while abandoning those for stakeholders? I will suggest the authors to more carefully diagnose their channels to explain this effect.

Response: We are grateful for the reviewer's thoughtful comments. To address the reviewer's concern about the lack of empirical evidence on the return on financial asset investments, we have conducted additional tests to examine the impact of the NAMR on the return on financial asset investments.

---

## [Decision Letter · Decision Letter 1]

18 Feb 2025

PONE-D-24-48995R1Financial Market Regulation and Corporate Social Responsibility: Evidence from China’s New Asset Management RegulationPLOS ONE

Dear Dr. Wang,

Thank you for submitting your manuscript to PLOS ONE. After careful consideration, we feel that it has merit but does not fully meet PLOS ONE’s publication criteria as it currently stands. Therefore, we invite you to submit a revised version of the manuscript that addresses the points raised during the review process.

Dear authors,

Please address the comments of the reviewers in detail. In addition, please pay attention to the concepts of ESG and CSR and try to address these concepts in detail.

We look forward to receiving your revised manuscript.

Kind regards,

Saddam A. Hazaea, Postdoctoral

Academic Editor

PLOS ONE

Journal Requirements:

Additional Editor Comments:

Dear authors,

Please address the comments of the reviewers in detail. In addition, please pay attention to the concepts of ESG and CSR and try to address these concepts in detail.

Reviewers' comments:

Reviewer's Responses to Questions

**Comments to the Author**

1. If the authors have adequately addressed your comments raised in a previous round of review and you feel that this manuscript is now acceptable for publication, you may indicate that here to bypass the “Comments to the Author” section, enter your conflict of interest statement in the “Confidential to Editor” section, and submit your "Accept" recommendation.

Reviewer #2: All comments have been addressed

Reviewer #3: All comments have been addressed

Reviewer #4: (No Response)

2. Is the manuscript technically sound, and do the data support the conclusions?

Reviewer #2: Partly

Reviewer #3: Yes

Reviewer #4: Yes

3. Has the statistical analysis been performed appropriately and rigorously? 

Reviewer #2: Yes

Reviewer #3: Yes

Reviewer #4: Yes

4. Have the authors made all data underlying the findings in their manuscript fully available?

Reviewer #2: Yes

Reviewer #3: Yes

Reviewer #4: (No Response)

5. Is the manuscript presented in an intelligible fashion and written in standard English?

Reviewer #2: Yes

Reviewer #3: Yes

Reviewer #4: (No Response)

6. Review Comments to the Author

Reviewer #2: I appreciate the authors for their responses to my comments. Most of the comments are well explained. Some additional improvements can be made before this paper gets published.

1. There are still no captions for each table and figure.

2. The paper writing should be further polished. There are frequent typos and grammar errors in the paper. Additionally, the introduction and conclusion can be more concise while emphasizing the contribution of this paper to the existing literature.

Reviewer #3: (No Response)

Reviewer #4: 1.ESG and CSR concepts are highly correlated. So, the author should explain the correlation between the two concepts in the variable description. And explain the differences between ESG and CSR.

2.ESG encompasses corporate performance across environmental, social, and governance dimensions, with the social component being most closely associated with CSR. However, the results presented in Table 10 indicate that NAMR demonstrates no significant impact on social performance. The author needs to provide an explanation for this finding.

7. PLOS authors have the option to publish the peer review history of their article (what does this mean? ). If published, this will include your full peer review and any attached files.

**Do you want your identity to be public for this peer review?** For information about this choice, including consent withdrawal, please see our Privacy Policy .

Reviewer #2: No

Reviewer #3: No

Reviewer #4: No

---

## [Author Response · Author response to Decision Letter 2]

1 Apr 2025

Response to reviewers

Dear editor and reviewers,

Thank you for offering us an opportunity to improve the quality of our submitted manuscript (Financial Market Regulation and Corporate Social Responsibility: Evidence from China's New Asset Management Regulation Manuscript. Number.: PONE-D-24-48995R1). We appreciated very much the reviewers' constructive and insightful comments. In this revision, we have addressed all of these comments. We hope the revised manuscript has now met the publication standard of your journal.

We highlighted all the revisions in red color.

On the following pages, our point-to-point responses to the queries raised by the reviewers are listed.

Suggestions from the editor

1.Dear authors, Please address the comments of the reviewers in detail. In addition, please pay attention to the concepts of ESG and CSR and try to address these concepts in detail.

Response: Thank you for your valuable reminder. Following the reviewers' suggestions, we have provided a detailed explanation of the correlation between ESG and CSR in the variable description section. Additionally, we have clarified the differences between ESG and CSR, and we have elaborated on the scientific and rational basis for using ESG as a measure for CSR.

2.Please review your reference list to ensure that it is complete and correct. If you have cited papers that have been retracted, please include the rationale for doing so in the manuscript text, or remove these references and replace them with relevant current references. Any changes to the reference list should be mentioned in the rebuttal letter that accompanies your revised manuscript. If you need to cite a retracted article, indicate the article’s retracted status in the References list and also include a citation and full reference for the retraction notice.

Response: Thank you for your reminder regarding the reference list. We have reviewed and updated the references in response to the first-round reviewer's suggestions, which recommended enriching the research background and emphasizing the contributions of our study. The newly added citations include:

1. Dong Y, Dong M, Tan S, Ge R. Shadow banking, financial regulation, and bank risk in China. Financ Res Lett. 2024;63: 105293. doi: https://doi.org/10.1016/j.frl.2024.105293

2. Jungo J, Madaleno M, Botelho A. The Effect of Financial Inclusion and Competitiveness on Financial Stability: Why Financial Regulation Matters in Developing Countries? Journal of Risk and Financial Management. 2022;15. doi:10.3390/jrfm15030122

3. Ullah W, Zubir ASM, Ariff AM. Exploring the moderating effect of regulatory quality on the relationship between financial development and economic growth/economic volatility for developed and developing countries. Borsa Istanbul Review. 2024;24: 934–944. doi: https://doi.org/10.1016/j.bir.2024.04.015

4. Delle Foglie A, Boukrami E, Vento G, Panetta IC. The regulators’ dilemma and the global banking regulation: the case of the dual financial systems. Journal of Banking Regulation. 2023;24. doi:10.1057/s41261-022-00196-2

5. Ofoeda I, Amoah L, Anarfo EB, Abor JY. Financial inclusion and economic growth: What roles do institutions and financial regulation play? International Journal of Finance and Economics. 2024;29. doi:10.1002/ijfe.2709

6. Liu Q, Wu J. Strong financial regulation and corporate risk-taking: Evidence from a natural experiment in China. Financ Res Lett. 2023;54. doi:10.1016/j.frl.2023.103747

7. Zhang B, Guo M. Strong financial regulation and the intelligent transformation of enterprises. Econ Anal Policy. 2025;85: 186–207. doi: https://doi.org/10.1016/j.eap.2024.11.017

8. Jiang M, Zhou W, Song Y. Research on shadow banking, the new regulations on asset management and corporate finance. Stud of Int Financ. 2020;12: 63–72.

9. Liu C. Disruption of corporate financialization and labor cost growth: Evidence from China’s new asset management rules. PLoS One. 2023;18. doi:10.1371/journal.pone.0286683

In this revision, we have added new references to better explain the concepts of ESG and CSR and their interrelationship. These additional citations are as follows:

58 Carroll AB. The pyramid of corporate social responsibility: Toward the moral management of organizational stakeholders. Bus Horiz. 1991;34. doi:10.1016/0007-6813(91)90005-G

59. Yoon B, Lee JH, Byun R. Does ESG performance enhance firm value? Evidence from Korea. Sustainability (Switzerland). 2018;10. doi:10.3390/su10103635

60. Park JG, Park K, Noh H, Kim YG. Characterization of CSR, ESG, and Corporate Citizenship through a Text Mining-Based Review of Literature. Sustainability (Switzerland). 2023;15. doi:10.3390/su15053892

Reviewer #2

Comment 1. There are still no captions for each table and figure.

Response Thank you for your insightful comments. We have now added descriptive captions to every table and figure in the revised manuscript. the table now includes a note that summarizes its contents and explains the variables, and the figure has an accompanying caption that clearly outlines its data and significance.

Comment 2:The paper writing should be further polished. There are frequent typos and grammar errors in the paper. Additionally, the introduction and conclusion can be more concise while emphasizing the contribution of this paper to the existing literature.

Response Thank you for your valuable advice. We have thoroughly proofread the manuscript to correct typos and grammatical errors, ensuring a polished and professional presentation throughout. We have revised both the Introduction and Conclusion to be more concise. We believe these revisions enhance the clarity, flow, and overall impact of the manuscript. Thank you again for your constructive suggestions.

Reviewer #4

Comment 1:.ESG and CSR concepts are highly correlated. So, the author should explain the correlation between the two concepts in the variable description. And explain the differences between ESG and CSR.

Response Thank you for your valuable advice. We fully agree with your suggestion to clarify the correlation between ESG and CSR, as well as to explain the differences between the two concepts. In the revised manuscript, we have provided a more detailed distinction between the two concepts, as follows:

"The key dependent variable in this study is corporate social responsibility (CSR). CSR encompasses corporations' voluntary efforts to exceed legal and financial obligations, contributing to stakeholders through ethical practices, social initiatives, and sustainability efforts [58]. Yoon et al. describe CSR as a broad, discretionary concept that includes social, ethical, and environmental activities beyond regulatory requirements [59]. However, measuring CSR has been a persistent challenge due to its subjective and qualitative nature. ESG emerged as a response to this challenge by standardizing and quantifying corporate responsibility efforts, making them comparable and transparent across firms and industries. As a result, ESG is not separate from CSR but rather an institutionalized mechanism that translates CSR commitments into measurable environmental, social, and governance indicators. Park et al. conduct a text-mining analysis and found that CSR and ESG are frequently discussed together, but their conceptual focus differs: CSR is qualitative and voluntary, focusing on corporate goodwill and ethical obligations. ESG is quantifiable and investment-driven, providing a framework that integrates corporate responsibility into financial and risk assessments [60]. In other words, ESG serves as a mechanism to operationalize CSR, ensuring that corporate responsibility efforts are systematically assessed and financially material to investors.

Given the strong correlation between CSR and ESG, ESG may serve as a proxy for CSR, offering more systematic coverage and improved data availability. We utilize Bloomberg's ESG composite score. The Bloomberg ESG measure minimizes measurement mistakes by offering a consistent and thorough evaluation of CSR behaviors. The ESG score assesses the quality and comprehensiveness of a firm’s environmental and social governance disclosure and reporting activities and is based on several quantitative and policy-relevant indicators. Compared to other ESG ratings, the Bloomberg ESG composite score is more comprehensive and includes sub-scores for the E, S, and G dimensions. Therefore, the higher the ESG score, the greater a firm’s engagement in CSR activities. Furthermore, the extensive usage of Bloomberg CSR rating scores in the body of current CSR research lends credence to the data's legitimacy and authenticity [61–64]."

We have added the following references to the revised manuscript:

58 Carroll AB. The pyramid of corporate social responsibility: Toward the moral management of organizational stakeholders. Bus Horiz. 1991;34. doi:10.1016/0007-6813(91)90005-G

59. Yoon B, Lee JH, Byun R. Does ESG performance enhance firm value? Evidence from Korea. Sustainability (Switzerland). 2018;10. doi:10.3390/su10103635

60. Park JG, Park K, Noh H, Kim YG. Characterization of CSR, ESG, and Corporate Citizenship through a Text Mining-Based Review of Literature. Sustainability (Switzerland). 2023;15. doi:10.3390/su15053892

Once again, thank you for your insightful comments. We look forward to your further feedback.

Comment 2: ESG encompasses corporate performance across environmental, social, and governance dimensions, with the social component being most closely associated with CSR. However, the results presented in Table 10 indicate that NAMR demonstrates no significant impact on social performance. The author needs to provide an explanation for this finding.

Response Thank you for your insightful comment .We have provided the following explanations for this finding:

Since NAMR restricts speculative financial activities, it forces firms to allocate resources more efficiently, influencing their engagement with ESG-related CSR initiatives. The reduction in financial flexibility might lead firms to prioritize governance compliance (G) and environmental (E) initiatives over voluntary social contributions (S). Under an ESG-driven CSR approach, firms must balance financial discipline with sustainability—meaning that governance (G) and environmental (E) commitments become more directly measurable under ESG frameworks. However, social responsibility (S) efforts may remain voluntary, which could explain why NAMR's impact on the S-dimension of CSR is weaker.

Delayed Impact on Social Performance: Unlike environmental initiatives (which require immediate investment) and governance compliance (which is highly regulated), social responsibility investments often have longer adjustment periods. NAMR may trigger changes that manifest beyond the observed time frame�the NAMR may trigger changes that manifest outside the time frame of observation. To further explore this, we conducted an additional experiment where we regressed social performance (S) lagged by one period. The results show a significant negative effect of NAMR on lagged social performance (L.S) (p<0.1). This suggests that the overall non-significance of the social dimension may be due to the lagged policy effect of NAMR, rather than a failure of the theoretical mechanism. This theory suggests that changes may take time to fully appear in the data.

For the clarity and readability of the paper, we included the lagged item regression results in the response to the review rather than in the revised manuscript draft. The results are presented below:

(1) (2)

VARIABLES S L.S

Post*Prefin -.00019 -0.053***

(-0.01) (-4.18)

Size 0.393*** 0.297***

(12.12) (8.89)

Lev -0.074*** -0.091***

(-3.83) (-4.79)

ROA 0.014 -0.016

(1.07) (-1.31)

Cashflow 0.018* 0.013

(1.87) (1.53)

Growth -0.790* -0.079

(-1.79) (-0.19)

Board -0.040** -0.044***

(-2.44) (-2.84)

Indep 0.013 0.011

(0.95) (0.83)

Top10 -0.027 0.004

(-1.01) (0.17)

Balance 0.036** 0.007

(2.12) (0.44)

INST 0.066** 0.018

(2.32) (0.66)

TMTPay 0.029* 0.039**

(1.74) (2.56)

Separate 0.003* 0.003

(1.69) (1.50)

Constant -0.317*** -0.498***

(-13.40) (-26.38)

Observations 6,820 5,867

R-squared 0.374 0.443

FIRM FIXED EFFECT Yes Yes

Year Fixed Effect Yes Yes

t-statistics in parentheses

*** p<0.01, ** p<0.05, * p<0.1

---

## [Decision Letter · Decision Letter 2]

15 Apr 2025

Financial Market Regulation and Corporate Social Responsibility: Evidence from China’s New Asset Management Regulation

PONE-D-24-48995R2

Dear Dr. Yichuan Wang

We’re pleased to inform you that your manuscript has been judged scientifically suitable for publication and will be formally accepted for publication once it meets all outstanding technical requirements.

Kind regards,

Saddam A. Hazaea, Postdoctoral

Academic Editor

PLOS ONE

Additional Editor Comments (optional):

Reviewers' comments:

Reviewer's Responses to Questions

**Comments to the Author**

1. If the authors have adequately addressed your comments raised in a previous round of review and you feel that this manuscript is now acceptable for publication, you may indicate that here to bypass the “Comments to the Author” section, enter your conflict of interest statement in the “Confidential to Editor” section, and submit your "Accept" recommendation.

Reviewer #3: All comments have been addressed

Reviewer #4: (No Response)

2. Is the manuscript technically sound, and do the data support the conclusions?

Reviewer #3: Yes

Reviewer #4: (No Response)

3. Has the statistical analysis been performed appropriately and rigorously? 

Reviewer #3: Yes

Reviewer #4: (No Response)

4. Have the authors made all data underlying the findings in their manuscript fully available?

Reviewer #3: Yes

Reviewer #4: (No Response)

5. Is the manuscript presented in an intelligible fashion and written in standard English?

Reviewer #3: Yes

Reviewer #4: (No Response)

6. Review Comments to the Author

Reviewer #3: I would like to express my gratitude to the researchers for their diligent efforts in making all the required amendments efficiently, improving the quality of the paper. Thank you for your dedication and commitment.

Reviewer #4: The author has revised the paper as required. This article has theoretical and practical significance, and the research method is reasonable. So, I suggest accepting this paper.

7. PLOS authors have the option to publish the peer review history of their article (what does this mean? ). If published, this will include your full peer review and any attached files.

**Do you want your identity to be public for this peer review?** For information about this choice, including consent withdrawal, please see our Privacy Policy .

Reviewer #3: No

Reviewer #4: No

---

## [Editor Report · Acceptance letter]

PONE-D-24-48995R2

PLOS ONE

Dear Dr. Wang,

I'm pleased to inform you that your manuscript has been deemed suitable for publication in PLOS ONE. Congratulations! Your manuscript is now being handed over to our production team.

Kind regards,

on behalf of

Dr. Saddam A. Hazaea

Academic Editor

PLOS ONE